# DNA Methylation Inhibitor 5-Azacytidine Promotes Leaf Senescence in Pak Choi (*Brassica rapa* subsp. *chinensis*) by Regulating Senescence-Related Genes

Yuntong Li [1], Junzhen Zhu [2], Xiaoyang Xu [2], Pengxia Li [1,2,3,4,5,*] and Xuesong Liu [3,5]

1   Department of Food Science, Shenyang Agricultural University, Shenyang 110866, China
2   School of Food and Biological Engineering, Jiangsu University, Zhenjiang 212013, China
3   Institute of Agricultural Facilities and Equipment, Jiangsu Academy of Agricultural Sciences, Nanjing 210014, China
4   Jiangsu Key Laboratory for Horticultural Crop Genetic Improvement, Nanjing 210014, China
5   Key Laboratory of Cold Chain Logistics Technology for Agro-Products, Ministry of Agriculture and Rural Affairs, Nanjing 210014, China
*   Correspondence: guoshubaoxian@163.com

**Abstract:** Leaf senescence is strictly regulated by multiple internal factors and external environmental signals, with the epigenetic modification being an important element among them. However, the epigenetic mechanism of leaf senescence is largely unknown in horticultural crops, especially the leaf vegetable pak choi, which easily senesces, and becomes yellow post-harvest. In this study, we found that the expression of DNA methyltransferases (*BcMET1*, *BcSUVH4*, *BcDRM2*, *BcRDR2*, and *BcCMT3*) of pak choi decreased during storage. The preliminary results showed that its senescence process was accompanied by DNA methylation changes. Moreover, treatment with 500 μM 5-Azacytidine (5-Aza) (DNA methylation inhibitor) can promote the senescence of pak choi leaves by (1) increasing the degradation of chlorophyll (Chl) and its derivatives, (2) increasing the activities of Mg-dechelatase (MDCase), pheophytinase (PPH) and pheophorbide a oxygenase (PAO), and (3) inducing the expression of senescence-related genes (*BcSAG12*, *BcNYC1*, *BcSGR1*, *BcSGR2*, *BcPPH1*, *BcPPH2*, *BcPAO*, and *BcRCCR*), thereby accelerating the senescence of the pak choi leaves. Further studies showed that DNA demethylation occurred in the promoter regions of *BcSGR2* and *BcSAG12* during storage, with the bisulfite sequencing detection showing that their degrees of methylation decreased. Therefore, our findings help us understand how epigenetic modifications affect the storage tolerance of leafy vegetables, which is highly significant for cultivating anti-senescent vegetable varieties.

**Keywords:** pak choi; demethylation; 5-Aza; postharvest; senescence

## 1. Introduction

Pak choi (*Brassica rapa* subp. *chinensis*) is widely grown in southern China. It is a crisp and delicious vegetable rich in vitamins, carotene, carbohydrates, proteins, and other nutrients. However, it has tender leaves with a large leaf surface area which promotes vigorous respiration [1]. Therefore, wilting easily occurs during postharvest storage and transportation, which directly affects their shelf life and causes nutrient loss [2].

The yellowing of postharvest vegetable leaves is often accompanied by the degradation of macromolecules like Chl [3]. The pathway of Chl degradation has been elucidated using functional analysis of Chl catabolic genes [4]. Chl in higher plants mainly comprised chlorophyll a (Chl a) and chlorophyll b (Chl b) [5]. First, Chl b is reduced to 7-hydroxymethyl Chl a (HCA) under the action of NON-YELLOW-COLORING 1 (NYC1) and NYC1-LIKE (NOL), followed by the reduction of HCA to Chl a via 7-hydroxymethyl Chl a reductase (HCAR) [5,6]. Then the magnesium ion of Chl a is chelated via magnesium chelatase (MCS) to generate pheophytin a (Phein a) [7]. Under the action of pheophytinase (PPH),

Phein a transforms into pheophorbide a. Pheophorbide a is degraded into an unstable red Chl metabolite (RCC) via the PAO enzyme, which finally results in a colorless primary fluorescent Chl metabolite (PFCC) [8,9].

Current studies have found that DNA modifications and histone modifications are important in plant senescence [10,11]. In eukaryotes, DNA methylation is epigenetically regulated by an important form of DNA cytosine methylation which is often accompanied by gene silencing [12]. In plants, DNA methyltransferases mediate cytosine methylation. For example, MET1 (METHYLTRANSFERASE 1) mostly mediates CG methylation, while CMT3 and CMT2 mediate CHG methylation, whereas DRM2 directly mediates CHH methylation [13]. Studies have shown that since DNA methylation and demethylation are dynamic processes, some demethylation enzymes like ROS1, DME (DEMETER), DML2 (DEMETER-LIKE), and DML3 [14] may be involved. Although DNA methylation is a key epigenetic mechanism regulating gene expression, previous studies have found that histone-specific modifications of lysine or arginine are also involved in epigenetic regulation [15]. For example, histone H3K9me2 methylation is particularly important in DNA methylation, especially CHG methylation catalyzed by CMT3 [16]. Since DNA methylation and demethylation are vital in the transcriptional regulation of many genes, they help regulate many important life processes and inevitably affect plant senescence. DNA hypomethylation and developmental defects, including delay in leaf senescence, were due to the temporal and spatial downregulation of MET1 activity in Arabidopsis [17]. The transcriptomic analysis found that with increasing plant senescence, the expression levels of genes responsible for methylation (CMT3 and MET1) continued declining, whereas those of genes responsible for demethylation (DME, DML2, and DML3) continued increasing [18]. Methylation levels in the plant aerial parts were reduced when plants were senesced [19]. Recent studies have also found that the DML3 gene knockout in *Arabidopsis thaliana* increased the genomic DNA methylation level, thereby inhibiting the expression of senescence-associated genes (SAGs), which delayed leaf senescence [20]. However, the epigenetic mechanism of postharvest leaf senescence in pak choi is still unclear.

Therefore, since the DNA methylation inhibitor 5-Aza promoted postharvest pak choi leaf senescence via the Chl degradation pathway, we systematically elucidated the epigenetic mechanism of senescence in pak choi based on the changes in DNA methylation levels of senescence-related genes. Meanwhile, this study provides some relevant insights into the molecular mechanism of postharvest senescence in pak choi.

## 2. Materials and Methods

### 2.1. Plant Materials and Treatment

Pak chois were obtained from the experimental field of Jiangsu Academy of Agricultural Sciences, China. After 40 days (d) of sowing, pak choi with uniform size and no mechanical damage to its leaves was harvested; the samples were quickly transported to the laboratory. Uniformly sized pak chois were randomly divided into five groups, with 12 in each group. Pak choi was sprayed evenly with the different concentrations of DNA methylation inhibitor 5-Aza ($C_8H_{12}N_4O_5$, Sigma-Aldrich, St. Louis, MO, USA) (62.5 μM, 125 μM, 250 μM, and 500 μM). The control group was sprayed with distilled water. Each group was sprayed with about 100 mL solution and dried at $20 \pm 1$ °C. The processed pak chois were transferred to porous polyethylene food bags, with each bag containing 4 pak chois, and each treatment containing three replicates. These were then stored in the dark at 20 °C (relative humidity 80%) for 4 d. Leaves with their primary veins removed from 12 vegetables in each treatment group were randomly selected and the samples were snap frozen in liquid nitrogen and then stored at −80 °C for determination of Chl content in leaves. The optimal concentration of 5-Aza was 500 μM according to the phenotype and total Chl content of pak choi leaves. Uniformly sized pak chois with no surface damage were randomly divided into the treatment and control groups, with 60 in each group. One group was evenly sprayed with 120 mL distilled water containing 120 μL tween (control group), while the other group was evenly sprayed with 120 mL 5-Aza (500 μM) containing

120 μL tween (treatment group), and air dried at $20 \pm 1$ °C. These pak chois were placed in porous polyethylene food bags, with four per bag, and three replicates per treatment per day. These were then stored at $20 \pm 1$ °C (relative humidity 80%) for 0, 1, 2, 3, 4 d. The leaves of the treatment and control groups without their primary vein were randomly selected every day, quickly froze in liquid nitrogen, and stored at $-80$ °C.

### 2.2. Surface Color and Chl Analyses

The CR-400 automatic Chroma Meter (Konica Minolta Sensing Americas, Inc., Ramsey, NJ, USA) was used to measure the color aberration of the pak choi leaves. Chlorophyll content was determined according to the method of Gao et al. [21]. The rapidly ground samples (0.2 g) were thoroughly homogenized with 10 mL of 80% acetone and incubated for 6 h at room temperature in the dark. The filtrate was extracted and the absorbance of the supernatant was measured at 642 nm and 665 nm using 80% acetone as a blank.

### 2.3. Chl-Degrading Enzyme Activity

Pak choi leaves (0.5 g) were homogenized in 0.05 M phosphate buffer (pH 7.4, 2 mL) containing 2.4 g $L^{-1}$ Triton X-100 (solarbio, Beijing, China) to prepare a crude enzyme extract. The suspension was thoroughly mixed and centrifuged at $10,000 \times g$ for 20 min at 4 °C. The resulting supernatant was used for subsequent experiments. The activities of MDCase, PPH, and PAO were determined as per the manufacturer's instructions of Abmart Kits (Abmart Shanghai Co., Ltd., Shanghai, China) for plant enzyme activity determination.

### 2.4. Chl-Degradation Derivative Content

The sample preparation was modified as described previously [1]. To 0.5 g sample, 10 mL pre-cooled acetone was added. Ultrasound treatment was subsequently carried out at 4 °C until the sample tissue was colorless and then centrifuged at $10,000 \times g$ for 20 min at 4 °C. Five milliliters of the supernatant were transferred into a clean 50 mL tube and 5 mL of 10% (*w/v*) NaCl was added to it. Then 10 mL of ether was added for extraction, followed by vigorous shaking, and the lower layer was discarded after stratification. Then 10 mL of pure water was added to the upper layer for washing, and it was subsequently washed twice. It was then shaken gently and the aqueous phase, left after stratification, was discarded. Anhydrous sodium sulfate (1 g) was added to absorb water from the sample, which was then slowly dried with a nitrogen blower (KY-II, Beijing, China). After blow-drying, 2 mL of pre-cooled acetone was added and left for 20 min to completely dissolve, followed by filtration through a 0.22 μm membrane. This extract was analyzed on an Agilent 1260 high-performance liquid chromatography-mass spectrometry (HPLC-MS) system (Agilent, Santa Clara, CA, USA). Chl derivative standards Pheide a, Phy a, Chlide a, and Chlide b were determined according to Dissanayake et al. [22].

### 2.5. RNA Isolation and Transcript Quantification

Trizol reagent (Invitrogen, Waltham, USA) was used to extract the total RNA from pak choi. According to the Thermo Kit instructions, cDNA was synthesized by reverse transcription. Quantitative reverse transcription polymerase chain reaction (qRT-PCR) used specific primers. Ubiquitin and Actin were used as the internal controls (Table 1). qRT-PCR was conducted in a 20 μL system, including 10 μL Supermix, 2 μL template cDNA, 1 μL forward and reverse primers (Table 1), and 6 μL sterile water. First, pre-denaturation was performed at 95 °C for 30 s, followed by 95 °C for 10 s, 60 °C for 30 s, and 40 cycles of reaction. Finally, a qPCR reaction was performed at 65 °C for 5 s and 95 °C for 5 s. Reactions were performed on a Biosystems 7500 Fast Real-Time PCR System (Applied Biosystems, Waltham, MA, USA) using an iTaq Universal SYBR Green Supermix (Takara TB Green® Fast qPCR Mix, Shiga, Japan). The expression of each gene was technically replicated thrice.

**Table 1.** Primer sequences used for PCR in this study.

| | Primer Name | Forward (5′–3′) | Reverse (5′–3′) |
|---|---|---|---|
| qRT-PCR | BcNYC1 | GGCTTGGTGGAGTTATCATTGG | ACGGATTCAGAACTGCGAGAT |
| | BcNOL | ACACAATCTATCGCCTGGAATG | AGATACTCAGCAACCACTTCAG |
| | BcSGR1 | AGCTTATTCAGACAAACGGGGT | TGGTGGATGCTTCTTGTCATCA |
| | BcSGR2 | ACAGTGACATAACCGCTAAGC | CTCCGCTAATGTGGCAATGAA |
| | BcPPH1 | GTGGTCGGTGAGAATGAGGA | CGCAGTGAGAAGTAGTGATTCG |
| | BcPPH2 | TGTGGTTGGTGAGAATGATGAC | TCGCAGTGAGAAGGAGTGATT |
| | BcPAO | TCTCTGAAGGAAGGTTGGATGA | TGAAGTAGCAGCCTGTGGAA |
| | BcRCCR | TCATCGTCAGTCACTCCTCAA | AACCTCAAGAACTTGCGTAGC |
| | BcSAG12 | CACTGGCGGCTTAACCACTGAA | GAAGATTGGCTGTATCCTACGGC |
| | BcMET1 | TGGTTTGGTTCTCGACGGAG | CGAGTTGTACAGTGCCCAGT |
| | BcSUVH4 | ATGATTGGTGACCTGCCAGG | ACACCAAGCCCTGAATCTCG |
| | BcDRM2 | TTCCAAACGAGCCAGGACTC | GACGCTCGGTCCTACTCATG |
| | BcRDR2 | AGAGCCTATGTTACGCCTTCA | TAGCCTTCCTTGGAGTTCACA |
| | BcCMT3 | GCAACTTGTTCGCTCAATCTC | GCCACTTCCGCTGTTACTC |
| | BcROS1 | TTTGCTGCAGGACTAGCTCC | GGTACTGGATACTCCGCAGC |
| | BcActin | TCTCTTCCACACGCCATCC | GTCTCCATCTCCTGCTCATAGT |
| | BcUbiquitin | GAGGTGGAGAGCAGTGACAC | GCTGTTTTCCGGCGAAGATC |
| Primers used for qPCR of enzyme digestion | BcSGR2-R1 | CTCCTTTACCCGAACCAACAAT | AGAGTACCCAATCTCCCTAACG |
| | BcSGR2-R2 | ATAGATAAGTTCCGACCGAAGC | GTAGCGTTGACGAGTTCTCTT |
| | BcSGR2-R3 | CACCTCGTCAGAGCGGATT | TTGTTGTTTGCGTGTTGGAGT |
| | BcSAG12-R1 | TACACCCATACATCAGCATTGT | TCAGATTCCAGTAGGCAAAGAT |
| | BcSAG12-R2 | GAAGAAGACTGACCAGCGATG | GAACAGACGAGCCGATCCT |
| | BcSAG12-R3 | TGAACCGAATAAACCGAATTGG | AAGCCCGAAGCACAAACTG |
| | BcRCCR-R1 | GGTCGGTGGAGAACATGGT | TTCATCTGCTCGGTCAAGAAC |
| | BcRCCR-R2 | AAGCACGGGATTAGATTTGGT | ACGGAAACTACCTACTAATTGC |
| | BcRCCR-R3 | CCAATTAAGTCGCTCTTGAGTC | TACACTAAACCGAACCCGTTAA |
| | BcNOL-R1 | AGACAGCAACCAAGAGGAACA | CTACCAACCTGGCAGATCAATG |
| | BcNOL-R2 | AGAGATGGCTGAGGCAAGG | GCTTCTTCCACACGCTTCC |
| | BcNOL-R3 | GTGAAGAGAAGAAGTTGATGGT | GAGCAGAAGATGAGGAACAGA |
| | BcNYC1-R1 | TTACTTCTCAGTGGTGCCTTCA | ATGGTTGCCTGCTGCTCTC |
| | BcNYC1-R2 | ACAGCTCTTGCGACCGTAG | CGTGGTGGTGTCTCTTGAATC |
| | BcNYC1-R3 | CGAGATGAGGTTGCCGTAAC | TCGGAGAAGGAAAGAGATGAGG |
| | BcPAO-R1 | CAGCGGGACTTAGGTTACAGA | GACCAGTTAAGCATCCAACAGT |
| | BcPAO-R2 | GGGTTTGGTTCTGATCGGTTT | CGCAAAGATCCAAATCGAACTC |
| | BcPAO-R3 | TGGATCGGTATCGGTTATGTTC | TGGCACTTGGCATAATAAGAAC |
| | BcPPH1-R1 | TGTAAGCAGCGTCCATAGAGA | TCCGTTCCTGAGCCTAAGC |
| | BcPPH1-R2 | TGTCATCGACCTGCTGAAGAA | CGGTGAGGATGCGATTGTTAT |
| | BcPPH1-R3 | TCTTTCCTCACCGTCCTTGTAA | TTCAGATTGCGGATGCTAGAAG |
| | BcPPH2-R1 | AATGGAAGGAGGAGGAGGATG | CACAGTTGACGGTTAGAGATTG |
| | BcPPH2-R2 | GCAACGGGTCTTTCAAATTGG | GCTGGCTTGGCTAACTTCTC |
| | BcPPH2-R3 | ACTTGGCTCTTACTGTCTGTGA | TTGTTAAGGTTGACGCACGAAT |
| | BcSGR1-R1 | TGATAACAGTGGACGGTCTTCT | GGTGGATGCGGTCATTGGA |
| | BcSGR1-R2 | TCTCTTCGAGTTTGCTCTGTTC | CAATCATACACCGTGACCTCAA |
| | BcSGR1-R3 | ACGCATCATCAGAAGAAGAACC | CTTTACCGAGGCTTGGAAACC |
| | BcActin | TCTCTTCCACACGCCATCC | GTCTCCATCTCCTGCTCATAGT |
| | BcUbiquitin | GAGGTGGAGAGCAGTGACA | CAAGGTACGACCGTCTTCAAG |
| Primers for disulfite qPCR | BcSGR2 | ATGTAGTGAAGAAGTTGGATATTAA | ATCAAAAACTAAAAACCCTTAAAAA |
| | BcSAG12 | AATTAATAGAGAAGAAGATTGATTAG | TAAACTAAATCAAATAAAAACAAAC |

### 2.6. DNA Methylation-Sensitive Restriction Enzyme Test

DNA methylation was detected by McrBc (methylation sensitive restriction endonuclease) enzyme digestion combined with qRT-PCR to extract the control group and treatment group of DNA at 0 d and 3 d. Then the McrBc restriction enzyme (New England Biolabs, Waltham, MA, USA) was used to digest each sample for 16 h. The reaction system was: 1 μL McrBc (10 U), 5 μL 10 × NEB Buffer, 0.5 μL 100 × GTP, 0.5 μL 100 × BSA, and 1 μg sample DNA, while the volume was adjusted with sterile water to 50 μL. It was then incubated at 37 °C for 16 h, followed by inactivation at 65 °C for 20 min, and finally diluted with sterile water to 100 μL after the experiment. The obtained enzyme digested product was used for detection of the expression level of each gene fragment in the sample via qRT-PCR. All results required at least two independent biological replicates and two independent digestion reactions. Quantitative results were obtained by normalized pak choi reference genes *Ubiquitin* and *Actin* (Table 1) and then compared with the DNA of undigested samples to calculate the ratio.

### 2.7. Bisulfite Sequencing Detection

According to Zhong et al. [23], the control group and treatment group of DNA at 0 d and 3 d were treated with the ZYMO EZ DNA methylation-Gold Kit (ZYMO Research, Waltham, MA, USA). The product was amplified by PCR, followed by cloning the purified target fragment into the vector (pEASY®-T1 Simple Cloning Kit, TransGen, Beijing, China) and then transformed into E. coli. The target gene methylation-specific primers (Table 1) were used for bacterial liquid PCR screening. Twenty positive clones of each of the two genes were selected, and sent to Tsingke Biotechnology Co., Ltd., (Beijing, China) for sequencing. BiQ Analyzer (http://biq-analyzer.bioinf.mpi-inf.mpg.de, accessed on 1 June 2021) was used to analyze the methylation level of the methylation sites after removing the sequences of the target fragments.

### 2.8. Statistical Analysis

A completely random design was used in this experiment. Three replicates were set for each treatment, with all data in this study being analyzed using SPSS 24.0 (version 24 for Windows, SPPS Inc., Chicago, IL, USA). One-way ANOVA and the LSD test were used for data in this experiment. Differences of the experimental results with * $p < 0.05$ and ** $p < 0.01$ were considered significant and extremely significant, respectively.

## 3. Results

### 3.1. Expression of DNA Methyltransferase

Promoter hypermethylation of SAGs in the Arabidopsis *atdml3* mutant causes delayed plant senescence, thus suggesting that DNA demethylation occurs during plant senescence [20]. For exploring the relationship between DNA methylation and senescence in pak choi, we detected the expression of DNA methyltransferase in pak choi. The expression of the methyltransferase genes, *BcMET1*, *BcSUVH4*, *BcDRM2*, *BcRDR2*, and *BcCMT3*, showed a downward trend with increasing storage period (Figure 1A–E). However, the expression of DNA demethylation related genes (*BcROS1*) remained unchanged (Figure 1F). Therefore, these results preliminarily suggest that the senescence of pak choi leaves may be closely related to the DNA methylation.

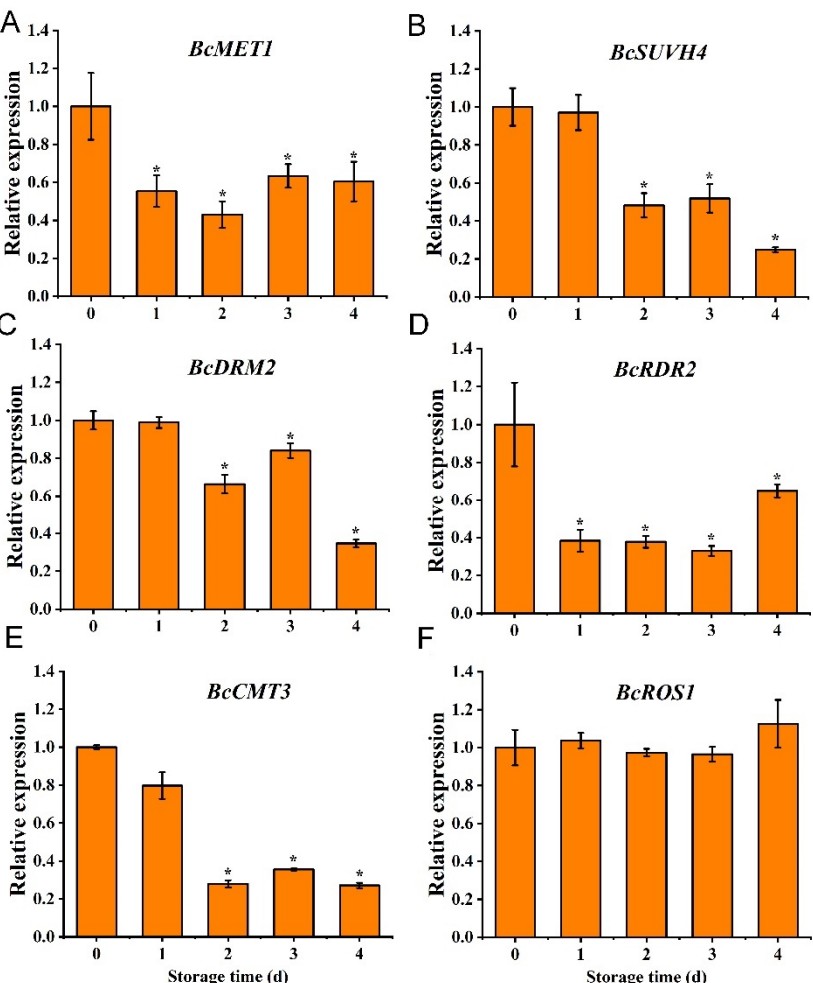

**Figure 1.** The expression of the DNA methyltransferase genes (*BcMET1*, *BcSUVH4*, *BcDRM2*, *BcRDR2*, and *BcCMT3*) (**A–E**) and the demethyltransferase gene (*BcROS1*) (**F**) in the control group during storage. The data in the figure are the mean ± SD of three biological replicates. The bar in the figure represents the standard deviation. The asterisks in the figure indicate significant differences between the mean values expressed for different storage days in the control group (* $p < 0.05$).

*3.2. Treatment with Different Concentrations of 5-Aza on the Phenotypic Characteristics and Total Chl Content*

5-Aza is a DNA methylation inhibitor, which can bind to the DNA methyltransferase to inhibit DNA methylation. Therefore, we used it to study the effect of DNA methylation on the senescence of pak choi leaves. We used different concentrations of 5-Aza (62.5 μM, 125 μM, 250 μM, and 500 μM) to treat pak choi, while we used sterile water as the control group. Although there was little change during the two days of storage, the leaves started yellowing on the third day. With the increasing storage period, the degree of leaf yellowing in control groups was similar to that in 62.5 μM and 125 μM of 5-Aza treatment groups. Compared with the control group, leaf yellowing degree in 250 μM and 500 μM treatment groups were higher, and the yellowing degree in the 500 μM treatment group was higher than that in 250 μM treatment group (Figure 2A).

The results showed that the total Chl content in the 500 μM treatment group on the fourth day of storage was the lowest (Figure 2B). Therefore, these results indicate that methylation inhibitor 5-Aza can accelerate the yellowing of pak choi leaves, in a concentration-dependent manner. Therefore, we selected the 500 μM of 5-Aza for analyzing how DNA methylation affects the senescence mechanism of pak choi leaves.

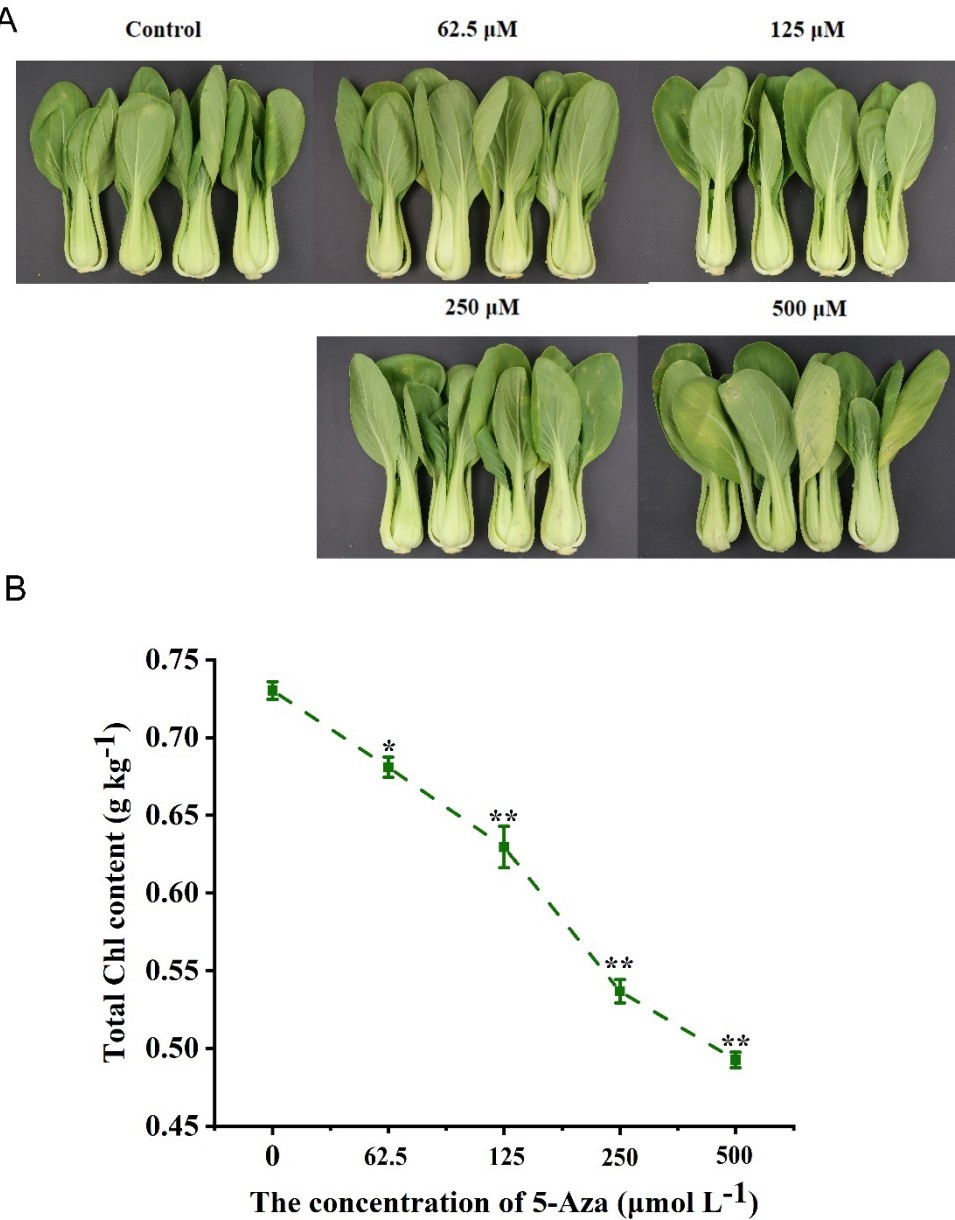

**Figure 2.** The phenotypic characteristics (**A**) and the changes of total Chl content (**B**) of pak choi treated with different concentrations of 5-Aza at $20 \pm 1$ °C. The data in the figure are the mean $\pm$ SD of three biological replicates. The asterisks in the figure indicate significant differences between the mean values of different concentrations (* $p < 0.05$; ** $p < 0.01$).

### 3.3. Surface Color and Total Chl Content

Yellowing of the leaf is the most noticeable symptom of senescence. The leaves of the control group turned yellow on day 4 of storage, as compared to the 5-Aza treatment group (500 μM) which started to yellowing on day 3, with most of the Chl being lost on day 4 (Figure 3A). The color difference index $L^*$ represents the leaf brightness, while $h^*$ represents leaf color. During storage, the $L^*$ values of both the control and 5-Aza treatment groups increased. On the third and fourth day of storage, the $L^*$ value of the 5-Aza treatment group was about 9.0% and 13.7% higher than the control group, respectively. The $h^*$ value during the storage period of the control and the treatment groups gradually decreased, with the $h^*$ value on days 3 and 4 of the 5-Aza treatment groups was about 3.5% and 5.0% lower than the control group, respectively (Figure 3B,C). Chl is also an important indicator of leaf senescence. Total Chl, Chl a, and Chl b also decreased gradually with the storage

time. On the third day of storage, Chl a, Chl b, and total Chl in 5-Aza treatment group were about 35.6%, 57.0%, and 36.2% lower than those in the control group, respectively. However, they decreased by 57.0%, 59.4%, and 57.2%, respectively, on the fourth day of storage (Figure 3D–F). It further shows that 5-Aza may promote the senescence of pak choi leaves.

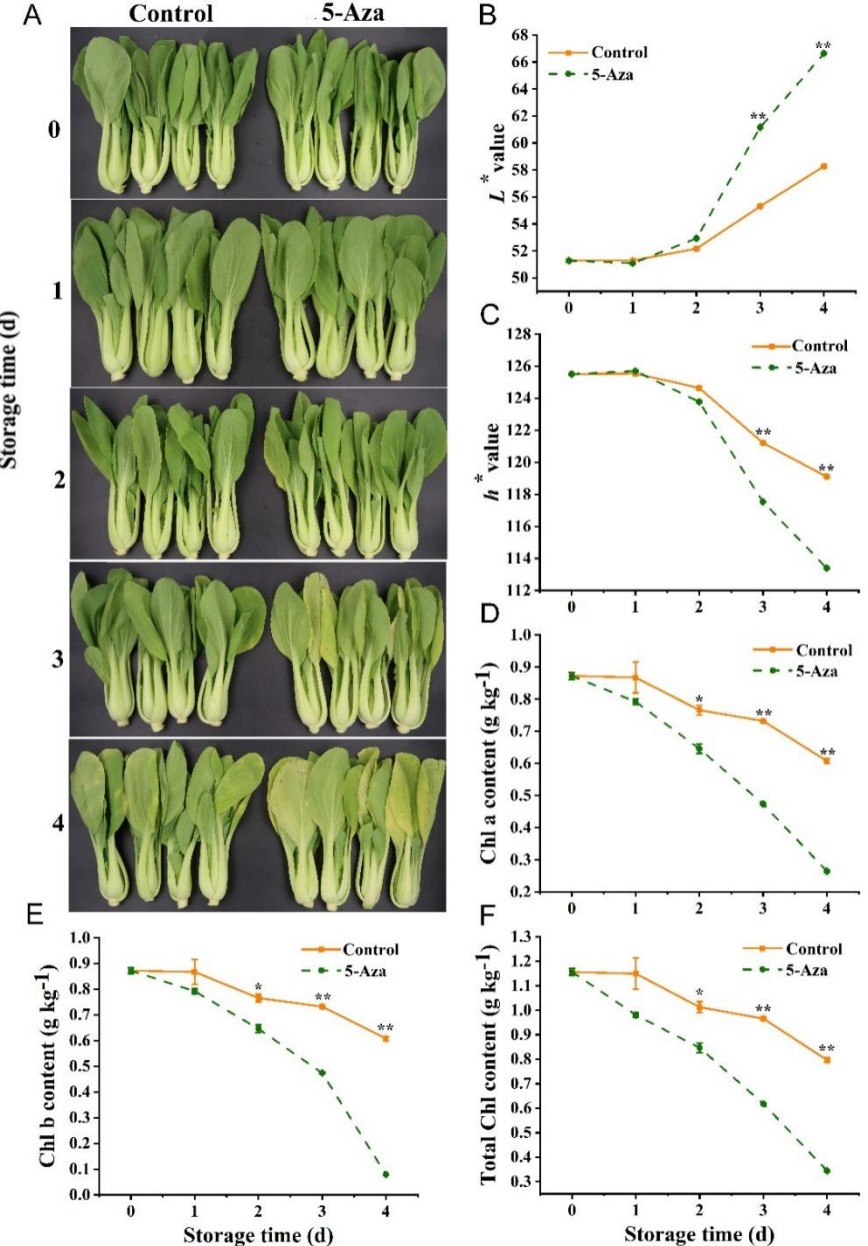

**Figure 3.** Phenotype (**A**), *L\** value (**B**), *h\** value (**C**), Chl a (**D**), Chl b (**E**) and the total Chl (**F**) of pak choi in the treatment and control groups at 20 ± 1 °C. The data in the figure are the mean ± SD of three biological replicates. The asterisks in the figure indicate significant differences between the mean values of treatments and controls (* $p < 0.05$; ** $p < 0.01$).

### 3.4. Activity of Chl-Degrading Enzymes

MDCase activity gradually increased during the first three days of storage, with the 5-Aza treatment group showing about 11.3%, 13.8%, and 18.2% higher activity than the control group, respectively (Figure 4A). PPH and PAO are two of the most important enzymes involved in Chl degradation. The PPH activity of the 5-Aza treatment group increased gradually during storage. On the third and fourth days of storage, the 5-Aza

treatment group showed about 9.8% and 9.0% higher activity than that of the control group (Figure 4B). However, PAO activity was higher in the middle and late stages of storage. On the second and fourth days of storage, the PAO activity of the 5-Aza treatment was about 29.2% and 37.4% higher than that of the control group, respectively (Figure 4C).

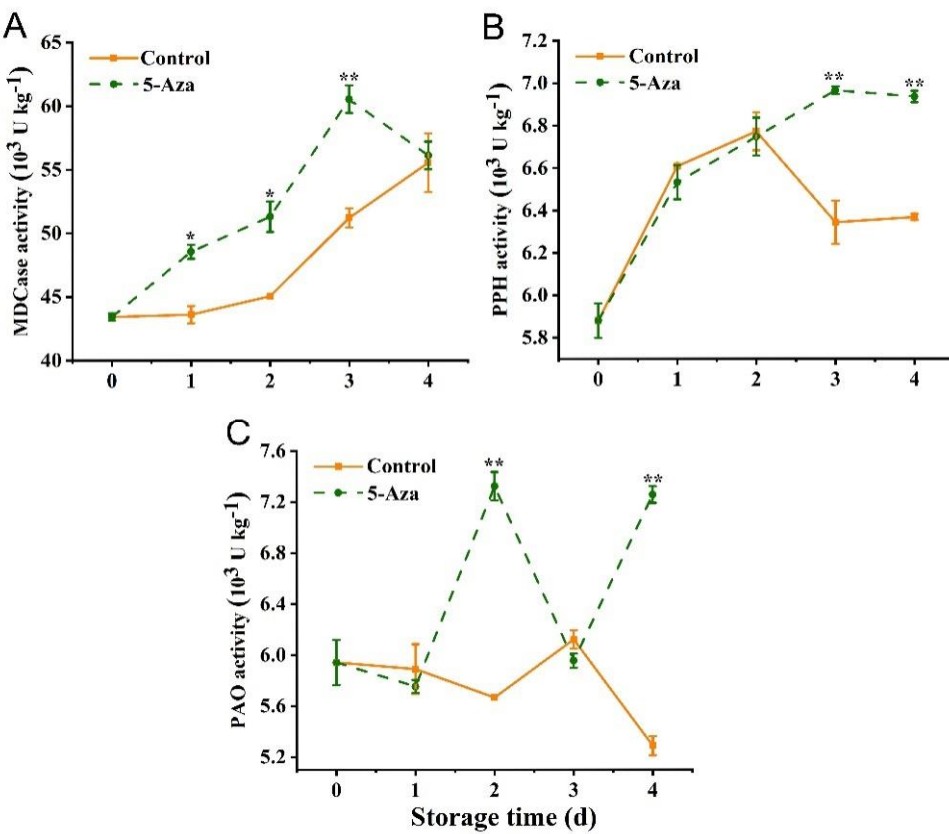

**Figure 4.** MDCase (**A**), PPH (**B**), and PAO (**C**) activities of pak choi in the treatment and control groups at $20 \pm 1$ °C. The data in the figure are the mean $\pm$ SD of three biological replicates. The asterisks in the figure indicate significant differences between the mean values of treatments and controls (* $p < 0.05$; ** $p < 0.01$).

### 3.5. Chl-Degradation Derivative Content

The Chl a and Chl b contents decreased gradually during storage. As compared with the control group, the Chl a content in the 5-Aza treatment group decreased by 26.2% and 46.1% on the third and fourth days, respectively (Figure 5A). Contrastingly, the 5-Aza treatment group showed about 45.1% and 61.1% lower Chl b content on the third and fourth days of storage, respectively (Figure 5B). During the storage period, the chlorophyllide (Chd) a content in the 5-Aza-treated leaves first decreased, then increased, and then again gradually decreased. However, those in the leaves of the control group decreased gradually. On the third and fourth day of storage, as compared with the control group, the Chd a content of the 5-Aza treatment reduced by about 12.4% and 36.1%, respectively (Figure 5C). The trend change of Chd b is similar to that of Chd a. After 3 and 4 d of storage, the Chd b content in the pak choi treated with 5-Aza was about 29.1% and 57.7% lower than that in the control group (Figure 5D). As shown in Figure 5E, the Phb a content in the treatment during storage was always lower than that in control group. The Phb a content in the treatment group was about 23.0%, 19.6%, and 35.3% lower than that of the control group on days 1, 3, and 4, respectively, but there was little difference in the content on 2 d. On the first day of storage, the Pheo a content in both the treatment and the control groups showed a decreasing trend. By comparing with the control group, the Pheo a content in the treatment group did not change (Figure 5F).

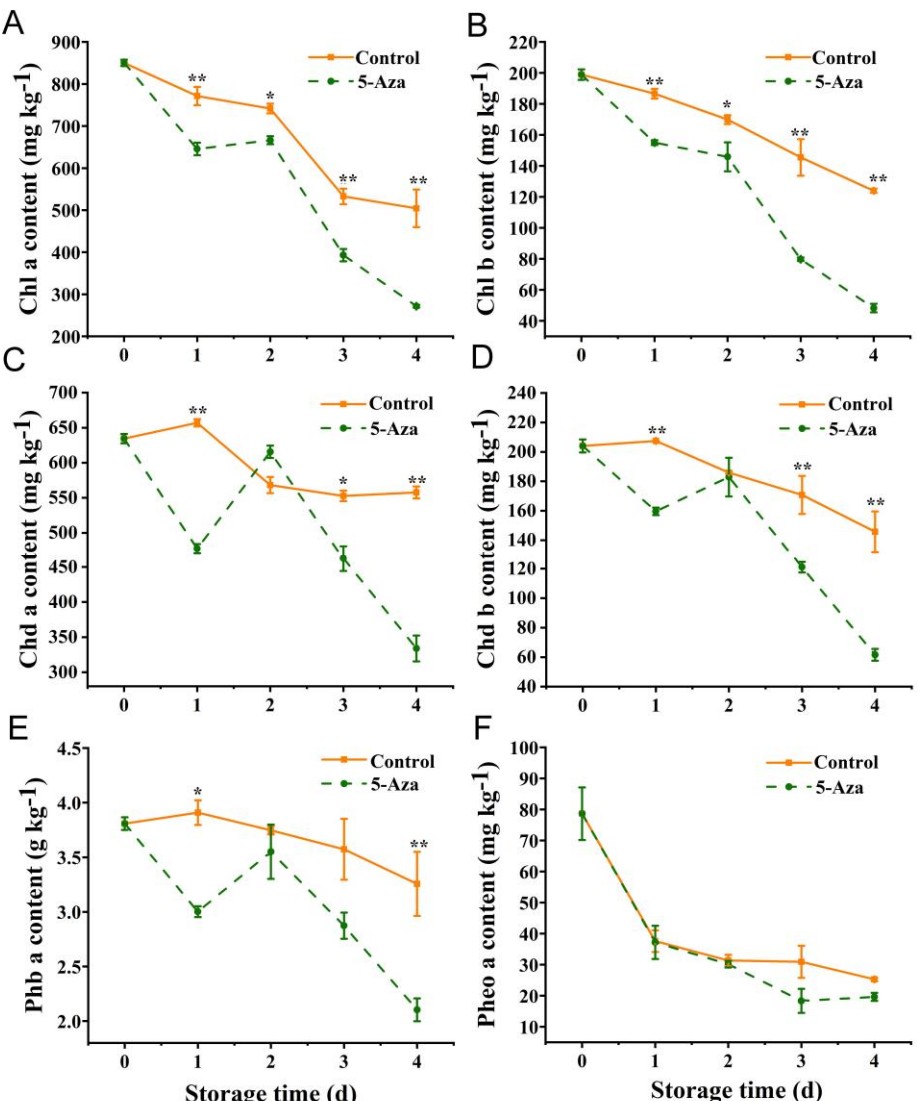

**Figure 5.** Chl a (**A**), Chl b (**B**), Chd a (**C**), Chd b (**D**), Phb a (**E**), Pheo a (**F**) contents of pak choi in the treatment and control groups at $20 \pm 1$ °C. The data in the figure are the mean $\pm$ SD of three biological replicates. The asterisks in the figure indicate significant differences between the mean values of treatments and controls (* $p < 0.05$; ** $p < 0.01$).

### 3.6. Expression of Senescence-Related Genes

Leaf senescence is closely related to the expression of senescence-related genes (Chl degradation and senescence marker genes). We used qRT-PCR to analyze the expression level changes of Chl catabolism genes during storage. The expression of *BcNYC1* increased on the first, second, and fourth days of storage. After the 5-Aza treatment, its expression level on the third and fourth day of storage was about 1.7 and 2.3 times that of the control group, respectively (Figure 6A). *BcNOL* expression increased during the first and fourth days of storage (Figure 6B). With increasing storage period, the expression of both *BcSGR1* and *BcSGR2* in both the control and the treatment groups was gradually up-regulated. The *BcSGR1* expression on the third and fourth days of storage was about 1.6 and 1.5 times higher than that of the control group, respectively (Figure 6C). Furthermore, on the third day of storage, the expression of *BcSGR2* in the treatment group was about 3 times higher than in the control group (Figure 6D). The expression of *BcPPH1* in the 5-Aza treatment group was about 1.5 and 1.4 times of that in the control group on the third and fourth day of storage, respectively (Figure 6E). With increasing storage time, the *BcPPH2* expression increased slightly, with those in the treatment group being about 3 and 1.3 times than in

the control group on the third and fourth days of storage, respectively (Figure 6F). The *BcPAO* expression increased slightly with increasing storage time, which increased after 5-Aza treatment as compared with the control group (Figure 6G). As shown in Figure 6H, the *BcRCCR* expression decreased with the increasing storage time, with it being about 3.4 times in the 5-Aza treatment group than in the control group on the third day of storage. The expression pattern of *BcSAG12* was similar to that of *BcSGR1* during storage. The expression level of *BcSAG12* in the treatment group was about 12.7 and 3.7 times of that in the control group on the third and fourth days of storage, respectively (Figure 6I).

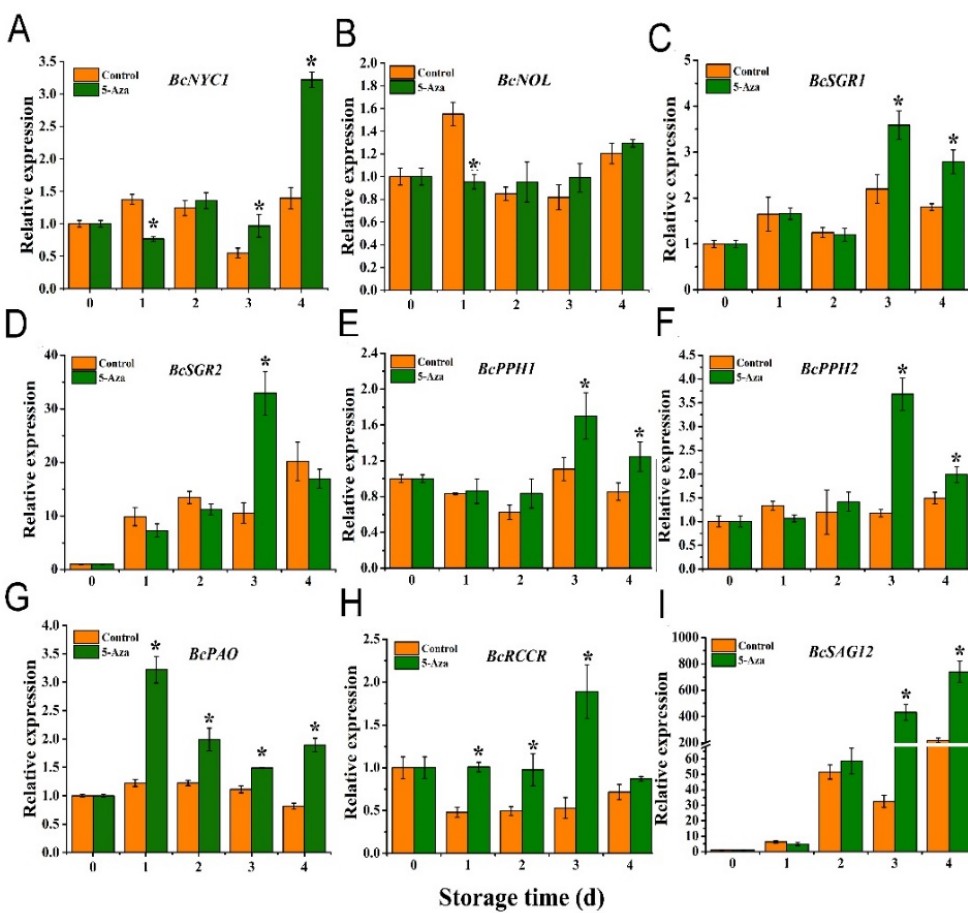

**Figure 6.** The detected expression levels of the eight Chl degrading genes (*BcNYC1*, *BcNOL*, *BcSGR1*, *BcSGR2*, *BcPPH1*, *BcPPH2*, *BcPAO*, and *BcRCCR*) (**A–H**) and the senescence marker gene (*BcSAG12*) (**I**) in pak choiat $20 \pm 1$ °C. The data in the figure are the mean $\pm$ SD of three biological replicates. The bar in the figure represents the standard deviation. The asterisks in the figure indicate significant differences between the mean values of treatments and controls (* $p < 0.05$).

### 3.7. DNA Methylation of Senescence-Related Genes

To explore the relationship between the expression level changes of senescence-related genes and DNA methylation, we further studied the genes (*BcNYC1*, *BcSGR1/2*, *BcPPH1/2*, *BcPAO*, *BcRCCR*, and *BcSAG12*) whose expression were induced by DNA methylation inhibitors. Methylation changes in the promoter regions of the above eight genes were preliminarily verified by the methylation sensitive restriction endonuclease McrBC method. We designed three pairs of methylation specific primers (R1, R2, R3) according to the CpG enriched regions of the *BcPPH1/2*, *BcSGR1/2*, *BcNYC1*, *BcPAO*, *BcRCCR*, and *BcSAG12* promoters on MethPrimer online website (http://www.urogene.org/cgi-bin/methprimer/methprimer.cgi, accessed on 25 June 2021). According to their phenotypic characteristics, we found that pak choi leaves started yellowing from the third day of storage. Therefore, we selected the DNA of pak choi of the control group stored on the 0 d and 3 d for subsequent

experimental verification. The results showed that the promoter region of *BcSGR2* was methylated at 0 d, while DNA demethylation occurred in the R3 region after 3 d of storage (Figure 7A). The promoter region of *BcSAG12* was highly methylated, with the DNA demethylation also occurring in the R3 region after storage for 3 d (Figure 7B). However, there was little change in the methylation of the promoter regions of *BcPPH1/2*, *BcNYC1*, *BcPAO*, *BcRCCR*, and *BcSGR1* (Figure 8A–F). In order to further verify the reliability of McrBC method, we used disulfite PCR to further verify the results. The results showed that the degree of methylation of the R3 region of *BcSGR2* promoter was 23.07% at 0 d and 11.53% after 3 d of storage. The degree of methylation of the treated group was 2.00% after 3 d of storage (Figure 7A). However, the degree of methylation of the R3 region of the *BcSAG12* promoter was about 31.03% at 0 d and 10.34% at 3 d after storage. The degree of methylation of the treated group was 3.25% after 3 d of storage (Figure 7B). Therefore, these results suggest that the postharvest pak choi may control their gene expression by reducing methylation in the SAGs promoter region, thus regulating leaf senescence.

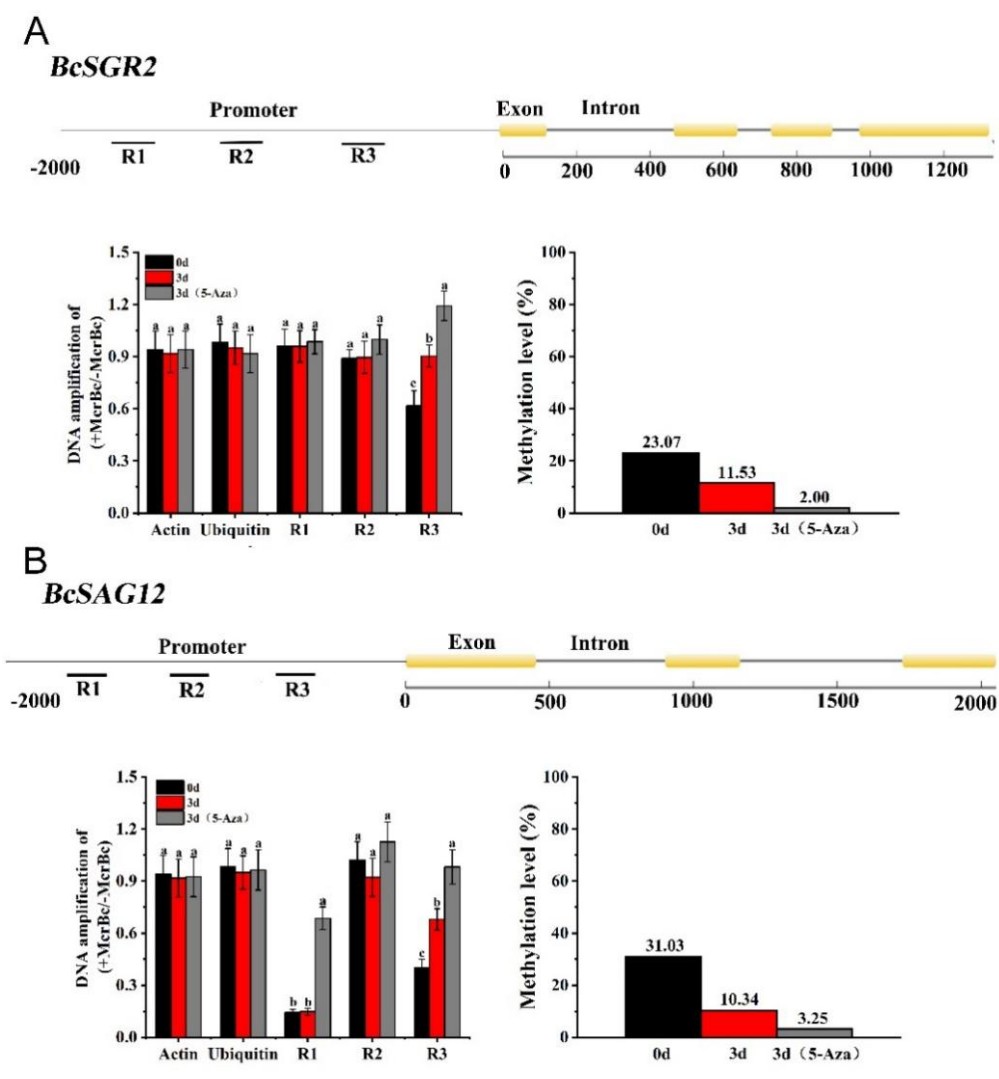

**Figure 7.** The degree of methylation of the Chl degradation gene (*BcSGR2*) (**A**) and the senescence gene (*BcSAG12*) (**B**) in the promoter regions R1, R2, and R3, and the methylation levels of *BcSGR2* (**A**) and *BcSAG12* (**B**) in the promoter R3 region at 0 d, 3 d, and 3 d (5-Aza) of storage. The data in the figure are the mean $\pm$ SD of three biological replicates. Different letters means statistically significant difference in the same same gene region ($p < 0.05$).

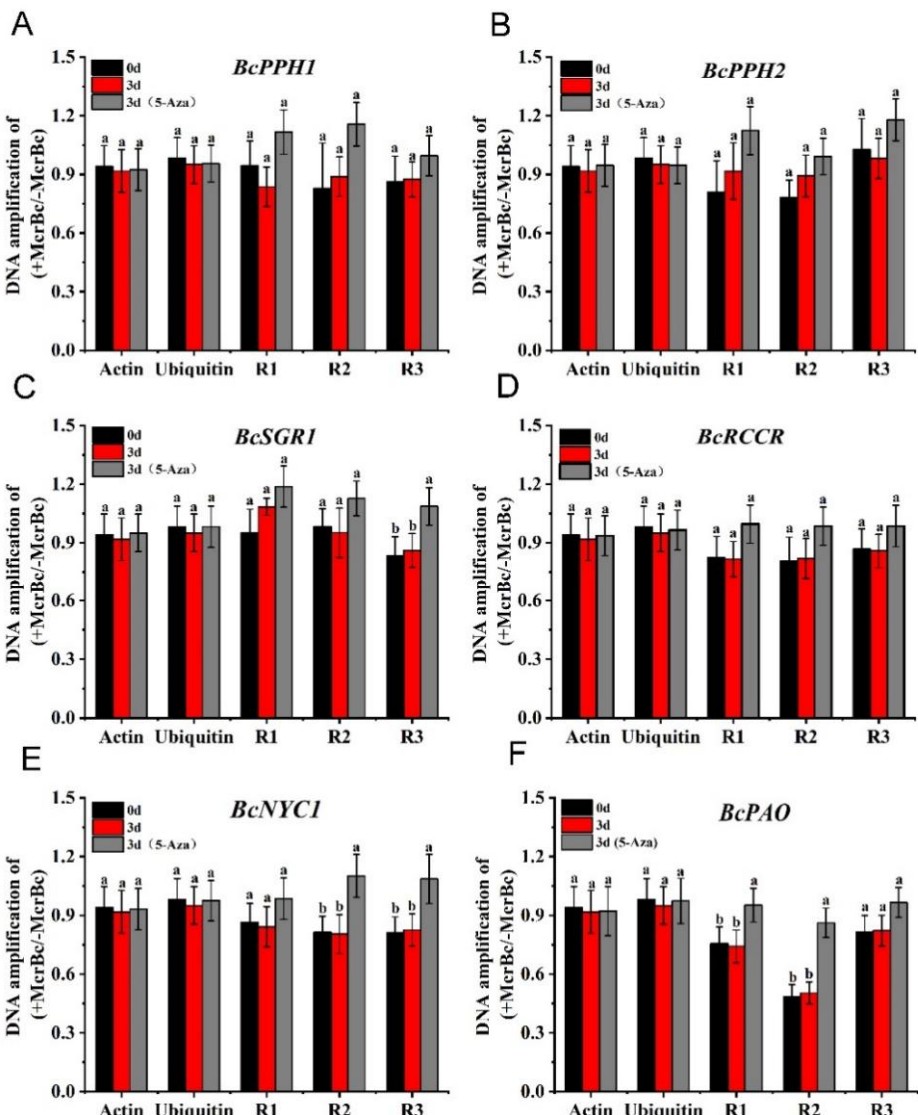

**Figure 8.** The methylation levels of the chlorophyll degradation genes (*BcPPH1*, *BcPPH2*, *BcSGR1*, *BcRCCR*, *BcNYC1*, and *BcPAO*) in the promoter regions R1, R2, and R3, at 0 d, 3 d, and 3 d (5-Aza) of storage (**A–F**). The data in the figure are the mean ± SD of three biological replicates. Different letters means statistically significant difference in the same same gene region ($p < 0.05$).

## 4. Discussion

DNA methylation/demethylation is an epigenetic mechanism that regulates genomic stability and gene expression, and is also involved in defense responses to environmental stresses [24,25]. In plants, dynamic methylation/demethylation occurs mainly on cytosine of the CG dinucleotide [16]. However, the current understanding of the epigenetic mechanisms associated with postharvest senescence (e.g., leaf yellowing) in pak choi is very limited. This study provided some evidence regarding the relationship between DNA demethylation, gene expression, and senescence in pak choi leaves after harvest.

Senescence is a critical stage in plant growth and development process, which requires simultaneous extensive reprogramming of the expression of diverse genes and multiple levels of regulation [26]. Studies have shown that besides gene reprogramming being a key factor in plant senescence, higher-level mechanisms like transcription factors and DNA methylation are also key factors in age-related genes [27–29]. In this paper, we explored the relationship between DNA methylation and postharvest senescence of pak choi. According to previous reports, the expression of DNA methyltransferase and DNA demethylase

decreased and increased in *Arabidopsis thaliana* during senescence [19]. The expression levels of DNA methyltranferases like *BcMET1*, *BcCMT3*, *BcSUVH4*, *BcDRM2*, and *BcRDR2* decreased with the increasing storage period (Figure 1A–E). DNA methylation is a dynamic process, the generation, maintenance, and demethylation are happening at any time [30]. However, we found that the demethyltransferase *BcROS1* expression level hardly changed with increasing storage time (Figure 1F). In conclusion, we can preliminarily speculate that postharvest leaf senescence of pak choi may be closely related to the level of genome-wide global DNA hypomethylation regulated by DNA methyltransferases.

To further study the relationship between postharvest senescence and DNA methylation of pak choi, 5-Aza, which is commonly used in plants, was used to inhibit the degree of DNA methylation. It is an analog of cytosine that irreversibly binds to DNA methyltransferases, thereby making it difficult for the genome to maintain a methylated state, thus causing DNA demethylation [23,31]. For example, 50 µM of 5-Aza reduced the level of DNA methylation in wheat genome and improved its salt tolerance [32]. To explore the optimal concentration of 5-Aza for affecting the senescence of postharvest pak choi, we used different concentrations of 5-Aza to treat postharvest pak choi. Compared with the control group, the leaves in the 500 µM 5-Aza treatment group showed the highest degree of yellowing, along with the lowest total Chl content during the entire storage process (Figure 2A,B). Therefore, we selected 500 µM of 5-Aza for follow-up experiments. We also found that on the third and fourth days of storage, the 5-Aza treatment group promoted the yellowing of pak choi leaves (Figure 3A). The surface color and Chl content also confirmed this conclusion (Figure 3B–F), thereby preliminarily confirming that the DNA methylation inhibitor 5-Aza may accelerate the senescence of pak choi leaves by reducing the DNA methylation level of their genome.

The degradation of Chl components during leaf senescence involves multiple Chl degradation and SAGs, including *BcNYC1*, *BcNOL*, *BcSGR*, *BcPAO*, *BcPPH*, *BcRCCR*, and *BcSAG12* [5]. SAG12 is a plant senescence-associated marker gene, which can largely reflect the process of plant senescence [33]. In this study, *BcSAG12* responded rapidly to senescence and 5-Aza exacerbated this phenomenon (Figure 6I). NYC1 and NOL are vital in converting Chl b to Chl a [34]. The expression of NYC1 in Chinese flowering cabbage was up-regulated during storage and also increased by MeJA [35]. We found that 5-Aza treatment promoted the increased expression of *BcNYC1* in pak choi (Figure 6A). The MDCase encoded by chloroplast SGR gene participates in Chl degradation by chelating $Mg^{2+}$ from Chl a [36]. The expression of *BcSGR1/2* in Chinese flowering cabbage was also found to be induced by MeJA [35]. Similarly, we observed that 5-Aza treatment up-regulated both the MD enzyme and *BcSGR1/2* expression (Figures 4A and 6C,D). The enzymes PPH and PAO hydrolyze Phy a into RCC [37]. Treatments like MeJA increased the expression of BcPPH in untreated Chinese flowering cabbage, thus accelerating senescence [38]. The expression pattern of *BcPPH1/2* also shared similarity with Tan et al. [36] (Figure 6E,F). PPH activity was also found to be promoted by 5-Aza (Figure 4B). The final two reactions of color loss in Chl were catalyzed by PAO and RCCR [37]. 5-Aza increased the expression of both *BcPAO* and *BcRCCR* (Figure 6G,H). Similar expression patterns of *BcPAO* and *BcRCCR* was observed in stored Chinese flowering cabbage post MeJA treatment [35]. 5-Aza also increased the activity of PAO (Figure 4C). Overall, with increasing storage period, the expression of the Chl degradation genes and the activity of Chl degrading enzymes were up-regulated. 5-Aza treatment increased these phenomena, thus promoting pak choi leaf senescence. As a result, 5-Aza reduced the accumulation of Chl a, Chl b, and Chl cycle intermediate metabolites like Chd a, Chd b, and Phb a (Figure 5A–E).

DNA methylation can regulate the expression of target genes, which varied with the location of DNA methylation [38,39]. DNA methylation in plants occurred in both the promoter and the transcribed gene ontology, with only 5% Arabidopsis genes being methylated in their promoter region, but many higher plants contain relatively large genomes and also have higher transposon content and more transposon adjacent genes. The methylation of the promoter region is more common, with the effect of methylation

changes on the whole plant being more noticeable [40,41]. DNA hypermethylation in promoter regions usually represses gene transcription [42–44]. Previous studies have found that DNA methylation levels gradually decreased during tomato ripening, whereas DNA methylation levels increased during citrus fruit growth [23,45]. It was found that during the cold storage of sweet orange, the methylation level of promoters of the anthocyanin biosynthesis-related genes, *DFR* and *Ruby*, decreased significantly in the high-pigmented (HP) areas and increased in not/low pigmented (NP) areas [46]. To clarify the relationship between postharvest leaf senescence and DNA methylation in pak choi, combined with phenotypic and physiological analysis, we analyzed the changes in DNA methylation in the promoter regions of senescence-related genes in pak choi at 0 and 3 d of storage period. We used McrBC digestion method in this study [47] to analyze the methylation patterns of senescence-related genes (*BcNYC1*, *BcSGR1/2*, *BcPPH1/2*, *BcPAO*, *BcRCCR*, and *BcSAG12*). The results showed that DNA demethylation occurred in the *BcSGR2* and *BcSAG12* promoter regions after 3 d of storage, which was further verified by the bisulfite sequencing (Figure 7A,B). Therefore, our results suggest that postharvest leaf senescence of pak choi may be mediated, at least partly, by DNA demethylation of senescence-related genes.

## 5. Conclusions

5-Aza treatment (1) promoted the degradation of Chl and its derivatives, (2) increased the activity of Chl degradation pathway related enzymes (MDCase, PPH, and PAO), (3) increased the expression of senescence related genes (*BcNYC1*, *BcPPH1/2*, *BcSGR1/2*, *BcPAO*, *BcRCCR*, and *BcSAG12*), and thereby promoted the senescence of postharvest pak choi. Additionally, McrBC digestion and bisulfite sequencing showed that the promoter of the Chl degradation related gene (*BcSGR2*) and senescence marker gene (*BcSAG12*) underwent DNA demethylation during the senescence process. In conclusion, the senescence of the postharvest pak choi may be caused by DNA demethylation of senescence-related genes.

**Author Contributions:** Y.L.; writing—original draft preparation, J.Z.; software, X.X.; methodology, P.L. and X.L.; writing—review and editing. All authors have read and agreed to the published version of the manuscript.

**Funding:** The work was supported by the National Natural Science Foundation of China (grant number 32001451, X.L.); the Key and General Programs of Modern Agriculture in Jiangsu Province (grant number BE2022368, P.L.).

**Data Availability Statement:** Data supporting reported results are available from the corresponding author on reasonable request.

**Acknowledgments:** The authors are grateful to Jialei Wang for their technical assistance with high-performance liquid chromatography-mass spectrometry.

**Conflicts of Interest:** The authors declare no known competing financial interests or personal relationships that could have appeared to influence the work reported in this paper.

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
