# Peer review of "DNA Methylation Inhibitor 5-Azacytidine Promotes Leaf Senescence in Pak Choi (Brassica rapa subsp. chinensis) by Regulating Senescence-Related Genes"

_agronomy, doi:10.3390/agronomy12102568_

Round 1
Reviewer 1 Report
In this manuscript, the authors investigated the epigenetic mechanism of pak choi's leaf senescence by researching the DNA methylation of chosen genes. Treatment with 500 µM 5-Azacytidine can hasten the senescence of pak choi leaves by increasing the degradation of chlorophyll and its derivatives, increasing the activities of Mg-dechelatase, pheophytinase, and pheophorbide an oxygenase, and inducing the expression of senescence-related genes. DNA demethylation of senescence-related genes may cause the postharvest pak choi's senescence.
The manuscript is well-structured and written with clear and concise scientific language. The title describes the contents of the manuscript. All parts of the manuscript (the abstract, the material and methods, the results, the conclusion, and the references) are adequately described.
The manuscript provided new knowledge about the function of DNA demethylation in the senescence of plants.
I recommend the article for publication.
I suggested a minor revision, as it is indicated in the attachment.

Author Response
In this manuscript, the authors investigated the epigenetic mechanism of pak choi's leaf senescence by researching the DNA methylation of chosen genes. Treatment with 500 µM 5-Azacytidine can hasten the senescence of pak choi leaves by increasing the degradation of chlorophyll and its derivatives, increasing the activities of Mg-dechelatase, pheophytinase, and pheophorbide an oxygenase, and inducing the expression of senescence-related genes. DNA demethylation of senescence-related genes may cause the postharvest pak choi's senescence.
The manuscript is well-structured and written with clear and concise scientific language. The title describes the contents of the manuscript. All parts of the manuscript (the abstract, the material and methods, the results, the conclusion, and the references) are adequately described.
The manuscript provided new knowledge about the function of DNA demethylation in the senescence of plants.
I recommend the article for publication.
I suggested a minor revision, as it is indicated in the attachment.
Response : Answer. Firstly, we greatly thank you for your approval of our research. Thank you very much for the USEFUL and IMPORTANT suggestions on our manuscript, which improve the quality of our manuscript. In the revised manuscript, we have corrected this error. Please check Line 176-177 and 191.
Reviewer 2 Report
The methodology used is probably improperly described as if the samples were frozen at the time of harvesting and then de-frozen and stored at 200C for 4 days (Lines 86-96) then most of the cells in these samples would be broken, with enzymatic activities from the programmed cell death cycle activated. This would severely damage both the Chlorophylls and the proteins studied as well as the genomic DNA integrity and methylation. I doubt the authors would have been able to obtain the results described in the manuscript under these conditions.
The description of the Chl content estimation seems hastily put together from previous reports of the team. First of all, it should be avoided during the Chl extraction to keep samples at room temperature for extended periods of time as the manuscript reports. The usual procedure includes quick grinding and acetone treatment (a few minutes), followed by centrifugation and collection of the supernatant. After a filtration (as described in the manuscript) no supernatant should be available as there should be no sediment. Thus a filtrate should be collected instead of supernatant.
Authors tend to excessively self-cite - out of 50 referenced materials almost 1/3 is from the authors. As DNA methylation is a broadly studied field by many teams, I find such a skew inappropriate.
Author Response
Firstly, we greatly thank you for your approval of our research. Thank you very much for the USEFUL and IMPORTANT suggestions on our manuscript, which improve the quality of our manuscript.
Point 1: The methodology used is probably improperly described as if the samples were frozen at the time of harvesting and then de-frozen and stored at 200C for 4 days (Lines 86-96) then most of the cells in these samples would be broken, with enzymatic activities from the programmed cell death cycle activated. This would severely damage both the Chlorophylls and the proteins studied as well as the genomic DNA integrity and methylation. I doubt the authors would have been able to obtain the results described in the manuscript under these conditions.
Response 1: Accepted. Thank you very much for your comment. We didn’t express it accurately enough; we have corrected this error. Please check Line 87-89.
Point 2: The description of the Chl content estimation seems hastily put together from previous reports of the team. First of all, it should be avoided during the Chl extraction to keep samples at room temperature for extended periods of time as the manuscript reports. The usual procedure includes quick grinding and acetone treatment (a few minutes), followed by centrifugation and collection of the supernatant. After a filtration (as described in the manuscript) no supernatantshould be available as there should be no sediment. Thus a filtrate should be collected instead of supernatant.
Response 2: Answer. Thank you very much for your question. The sample used in our experiment is already in the powder state after rapid grinding, and our expression is not accurate enough. We have corrected this mistake. We have corrected the statement of the collection supernatant. Chlorophyll content was determined according to the experimental method of Gao et al. Please check in Line 112-117.
Point 3: Authors tend to excessively self-cite - out of 50 referenced materials almost 1/3 is from the authors. As DNA methylation is a broadly studied field by many teams, I find such a skew inappropriate.
Response 3: Accepted. Sorry for this error. We have replaced the authors' references 23, 25, 32 . Please check in Line 578-580, 583-584 and 595-596.